# ITERATIVE WEAKLY SUPERVISED LEARNING FOR NOVEL CLASS OBJECT DETECTION

**Dejana Mandić[1], Wieland Brendel[2], Claudio Michaelis[1]**
[1] University of Tübingen, [2] Max Planck Institute for Intelligent Systems, Tübingen
`dejana.mandic@student.uni-tuebingen.de`

## ABSTRACT

Training object detectors for new classes usually requires collecting and labeling large amounts of data. Our paper introduces a new approach to address this issue - training novel-class object detectors using a combination of a few labeled images and weakly labeled data, that is easy to obtain. We propose an iterative fine-tuning framework that cycles through predicting pseudo-labels, filtering them using weak labels, and fine-tuning the model on this data. By repeating the process, we can mostly close the gap to a model trained on 40x more data, thereby offering a new approach to improving the trade-off between labeling effort and performance.

## 1 INTRODUCTION

Large-scale object detection datasets with comprehensive annotations have played a crucial role in advancing the field (Everingham et al., 2010; Lin et al., 2014; Cordts et al., 2016; Kuznetsova et al., 2020). However, detecting novel classes that are not in these datasets poses a significant challenge due to the high cost of annotating enough data. Although few-shot object detection methods (Kang et al., 2019; Huang et al., 2022) try to address this issue, they typically underperform compared to fully supervised methods. To optimize the trade-off between labeling effort and performance, we investigate a novel-class detection scenario that improves few-shot detection by combining weakly (Bilen & Vedaldi, 2016; Wang et al., 2022) and semi-supervised methods (Fang et al., 2021; Xiong et al., 2021). We propose an iterative fine-tuning framework that leverages a few fully labeled images and a larger set of weakly labeled images to learn and iteratively refine a detector's performance through multiple rounds of training and pseudo-label generation.

## 2 METHOD

**Weak Labels:** We consider two types of weak labels: (1) image-level labels, which are widely available from classification datasets or can easily be acquired in applications by recording additional images, and (2) point-class pairs, which only require clicking on each object once.

**Iterative Fine-tuning:** Figure 1 provides an overview of our proposed framework. It has three main steps: (1) fine-tune a pre-trained model using a small number of fully-labeled images; (2) generate pseudo-labels for the weakly labeled images by inferring bounding boxes and filtering them using the weak labels; (3) fine-tune the model using the pseudo-labeled dataset; (4) repeat parts (2) and

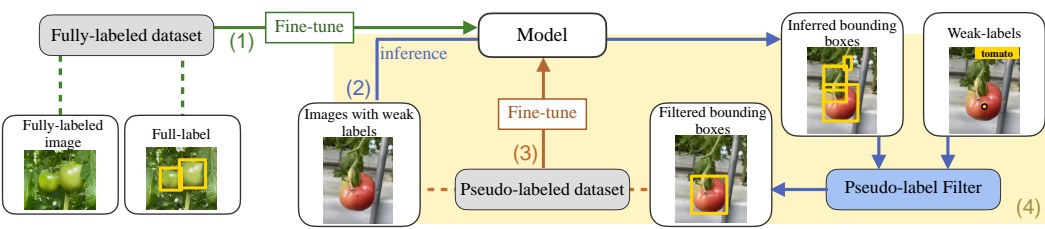

Figure 1: The iterative fine-tuning framework

(3) for a number of iterations. Note that we update the same model at each step, thus also improving the predicted pseudo-labels at each step.

**Pseudo-label Filter:** To filter inferred bounding boxes using weak labels, we build upon the approach by Wang et al. (2022). Given image-level class labels, we keep all bounding boxes with class probability higher than a threshold $\theta$. If there is no such bounding box, we keep the one with the highest probability. Given $N$ point-class pairs for an image, we filter the $K$ predicted bounding boxes using Hungarian matching and a loss function: $\mathcal{L}(p, c, b) = \gamma \cdot (1 - P(c)) + (1 - \gamma) \cdot d_2(p, b) \cdot \eta(p, b)$, where $\gamma \in [0, 1]$ is a trade-off, $p$ and $c$ are point and class labels, $b$ is a bounding box, $P(c)$ is a class probability, $d_2$ is the $L_2$ normalized distance between $p$ and the center of $b$. We perform min-max normalization across $N \times K$ distances. Finally, $\eta(p, b)$ is an indicator function that has value 1 if $p$ is inside $b$, otherwise equals infinity. Filtering examples are in Appendix A.2.

## 3 EXPERIMENTS

**Datasets:** For our experiments, we focus on the detection of tomatoes, which are not part of COCO (Lin et al., 2014) dataset we used for pretraining. The training and test data comes from Laboro-Tomato (LT)(Laboro.ai, 2020), a dataset with high-quality annotations for images of tomatoes in greenhouses. We create our own dataset split using half of the images for testing and the other half for fully and weakly labeled in distribution (IID) data. Additionally, we utilize tomato images from OpenImages (OI) (Kuznetsova et al., 2020) for weakly labeled out-of-distribution (OOD) data with image-level labels. Details of the label generation process are in Appendix A.1

**Model:** Following Omni-DETR (Wang et al., 2022), we use Deformable-DETR Zhu et al. (2020) as our model. However, the framework can easily be adapted to other object detection models.

**Experiments:** We compare training using 10 fully labeled images, with and without adding different types of weakly labeled data, with training on the full dataset of $\sim$400 images. We once determine all hyperparameters (threshold $\theta$, loss trade-off $\gamma$, and learning rate), and keep them constant for all experiments. We use the COCO $AP_{50:95}$ metric denoted as mAP (Lin et al., 2014).

## 4 RESULTS

Table 1 summarizes our findings. With the full dataset, Deformable-DETR achieves 69.8% mAP. With only 10 fully-labeled images, this drops to only 45.3%. But with weakly labeled data, this gap can mostly be closed. Simply adding 200 images with tomatoes from the existing Open Images dataset improves performance to 53.9%. Even better than these out-of-distribution (OOD) internet images is using in-distribution (IID) images from Laboro Tomato. With only 200 image-level labels we reach 59.9% and with point-class pairs, we reach 60.6% mAP. With more images ($\tilde{4}$00 in-

Table 1: Results

| # images | | weak | IID/ | iter- | |
| full | weak | label type | OOD | ations | mAP |
| --- | --- | --- | --- | --- | --- |
| 10 | - | - | - | - | 45.2 |
| 10 | 200 | image-level | OOD | 1 | 53.9 |
| 10 | 200 | image-level | IID | 1 | 59.9 |
| 10 | 200 | points | IID | 1 | **60.6** |
| 10 | 400 | points | IID | 1 | 61.9 |
| 10 | 400 | points | IID | 2 | 63.1 |
| 10 | 400 | points | IID | 3 | **63.2** |
| 10 | 400 | points | IID | 10 | 60.1 |
| 400 | - | - | - | - | 69.8 |

stead of 200) and multiple iterations, we reach 63.1%, mostly closing the gap while significantly reducing annotation effort. We find that the ideal number of iterations is 3. An excessive number of iterations leads to decreased performance because the model starts overfitting onto its own mistakes.

## 5 CONCLUSION

We here demonstrate that for a new class, the performance gap between using 400 or only 10 training images can mostly be closed by adding additional images with easy-to-obtain weak labels. Our proposed iterative fine-tuning framework provides an effective solution for developing single-class object detectors in low-data and limited annotation budget regimes. It is especially encouraging that repeated training iterations improve the accuracy of generated pseudo-labels and overall detector performance. In a next step, our framework could also be used to generate suggestions for human annotators, thus turning it into a tool to not only detect novel objects but also quickly and inexpensively label datasets.

URM STATEMENT

The authors acknowledge that at least one key author of this work meets the URM criteria of ICLR 2023 Tiny Papers Track.

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

## A APPENDIX

### A.1 ADDITIONAL INFORMATION ON OBTAINING DATA

**Obtaining image-level labels:** Since all images in Laboro Tomato include tomatoes, we can assign *tomato* as a label to all images from the dataset. For OpenImages, image-level labels can be obtained by getting all the object classes occurring in an image.

**Obtaining point-class pairs:** We sample points for point-class pairs similar to Wang et al. (2022) but with an improved sampling strategy. Wang et al. (2022) sample the points for the point-class labels anywhere within the segmentation mask. This way of sampling strongly affects our matching loss function, which is based on $L_2$ distance between the sampled point and the center of the bounding box, as the probability is very high that the sampled point will lie on the outer part of the segmentation mask. We want the sampled point to be near the ground truth bounding box center while allowing some noise to simulate human point annotations.

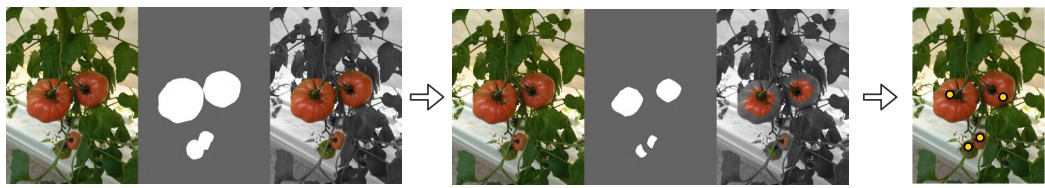

Figure 2: The process of sampling points from instance segmentation masks.

Figure 2 illustrates how we sample the points. The process goes as follows: (1) start from the ground truth segmentation mask; (2) use erode function from OpenCV library (Bradski, 2000) to narrow the segmentation mask; (3) sample the point from the narrowed segmentation mask. The kernel size used in the erode function is dependent on the area of the segmentation mask, so we get size-proportional narrowing.

**Label filtering:** The LaboroTomato dataset (Laboro.ai, 2020) contains multiple *tomato* subclasses, which were all unified to a single *tomato* class. OpenImages dataset (Kuznetsova et al., 2020), was filtered for tomato images in an outdoor environment, which resulted in $\sim 200$ images.

### A.2 PSEUDO-LABEL FILTERING EXAMPLES

Figure 3 shows examples of the pseudo-label filtering using weak labels and our pseudo-label filter: (left) image with image-level labels from OpenImages (OOD data); (right) image with point-class pairs from Laboro Tomato (IID data).

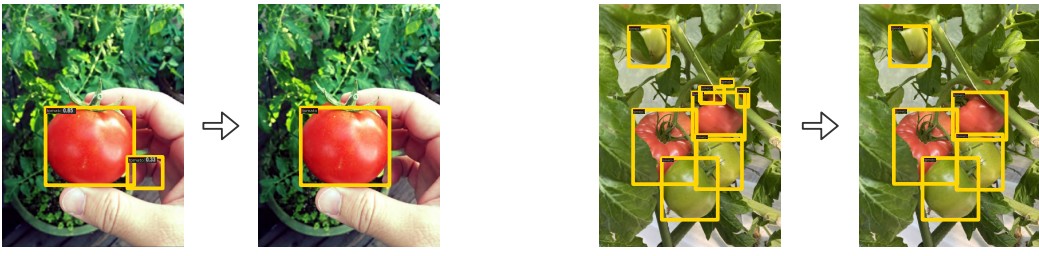

Figure 3: Examples of label filtering.

