# OpenReview forum: "Iterative weakly supervised learning for novel class object detection"
_ICLR.cc/2023/TinyPapers — Submitted to Tiny Papers @ ICLR 2023_

### Official Review · Reviewer_XxVW · 2023-03-26

**Confidence:** 5

**Summary Of Contributions:**

A framework for leveraging weakly labeled image to improve object detection on novel classes are proposed.

**Rating:**

Clear, Correct, and Reproducible (CCR): a submission which meets the reviewing criteria

**Strengths And Weaknesses:**

S1: I think overall this paper is very clear, especially in the experiments section.

S2: The paper also discusses in-depth on the implementation details and the used dataset, making it reproducible.

W1: the only experiment is on only one novel classes (tomatoes), It would be interesting and comprehensive to try the proposed framework on more classes.

W2: The comparison with other methods are lacking, for example, I think one baseline to compare to can be a semi-supervised object detection baseline which use the weakly labeled data as the unlabeled dataset.

**Suggested Changes:**

C1: Adding a comparison with semi-supervised object detection methods, and maybe another possible related detection method.

C2: I would suggest to extend the experiments on more classes other than just tomatoes.

---

### Official Review · Reviewer_4x6M · 2023-03-29

**Confidence:** 4

**Summary Of Contributions:**

This paper adapts the object detector for new category using a combination of a few labeled images and weakly labeled data. The proposed iterative fine-tuning framework cycles through predicting pseudo-labels, filtering them using weak labels, and fine-tuning the model on this data.

**Rating:**

Clear, Correct, and Reproducible (CCR): a submission which meets the reviewing criteria

**Strengths And Weaknesses:**

Strong aspects:

1. The paper addresses an important problem in object detection: detecting novel classes with limited labeled data.

2. The paper provides a clear overview of the proposed framework with a well-designed figure.

3. The proposed iterative fine-tuning framework can significantly reduce the labeling effort by using a few labeled images and weakly labeled data.

4. The paper includes detailed explanations of the weak label types used in the experiments and the pseudo-label filter method used to filter bounding boxes using weak labels.

**Suggested Changes:**

1. In Abstract, it is claimed that "training object detectors for new tasks". Regarding the setting in this paper, it might be more appropriate to use terms such as "for new classes" or "for new image domain".

---

### Author Response · Authors · 2023-05-31
**Revision and opt-in for archival**

We thank the reviewers for investing their time and effort to review our manuscript. We have uploaded the revised version with incorporated feedback and comments.

Additionally, we confirm that we wish to opt-in for archival and adhere that all the requirements are satisfied.

---

### Meta-Review · Area_Chair_cGBa · 2023-04-06

**Recommendation:** Invite to present
**Confidence:** 5

**Metareview:**

This paper proposed a framework for object detection of new categories using a few labeled examples and weakly labeled data.
The two reviewers both found the paper to be CCR, the method is presented clearly with detailed explanations of the implementation details.
The weaknesses found by reviewer are, the experiments are only limited to one class, and missing baselines in semi-supervised object detection.
Overall this paper is CCR, but it is recommended that the author to address some of the concerns by the reviewers in the final version.


**Summary:**

A clear method for weakly supervised object detection on novel class is proposed, the experimental details are clearly presented with effective performance.

**Reason For Not Giving A Higher Recommendation:**

The experiments only concerns one novel class, thus limits the validation of the effectiveness of the model.


**Reason For Not Giving A Lower Recommendation:**

Both reviewer agree that the paper is CCR

---

### Decision · Program_Chairs · 2023-04-08

Invite to present